# The Effect of Training on Localizing HoloLens-Generated 3D Sound Sources

**DOI:** 10.3390/s24113442

**Published:** 2024-05-27

**Authors:** Wonyeol Ryu, Sukhan Lee, Eunil Park

**Affiliations:** 1Department of Electrical and Computer Engineering, Sungkyunkwan University, Suwon 16419, Republic of Korea; ryuwon10@skku.edu; 2Artificial Intelligence Department, Sungkyunkwan University, Suwon 16419, Republic of Korea; 3Department of Intelligent Software, Sungkyunkwan University, Suwon 16419, Republic of Korea; pa1324@gmail.com

**Keywords:** VR, sound localization training, HRTF

## Abstract

Sound localization is a crucial aspect of human auditory perception. VR (virtual reality) technologies provide immersive audio platforms that allow human listeners to experience natural sounds based on their ability to localize sound. However, the simulations of sound generated by these platforms, which are based on the general head-related transfer function (HRTF), often lack accuracy in terms of individual sound perception and localization due to significant individual differences in this function. In this study, we aimed to investigate the disparities between the perceived locations of sound sources by users and the locations generated by the platform. Our goal was to determine if it is possible to train users to adapt to the platform-generated sound sources. We utilized the Microsoft HoloLens 2 virtual platform and collected data from 12 subjects based on six separate training sessions arranged in 2 weeks. We employed three modes of training to assess their effects on sound localization, in particular for studying the impacts of multimodal error, visual, and sound guidance in combination with kinesthetic/postural guidance, on the effectiveness of the training. We analyzed the collected data in terms of the training effect between pre- and post-sessions as well as the retention effect between two separate sessions based on subject-wise paired statistics. Our findings indicate that, as far as the training effect between pre- and post-sessions is concerned, the effect is proven to be statistically significant, in particular in the case wherein kinesthetic/postural guidance is mixed with visual and sound guidance. Conversely, visual error guidance alone was found to be largely ineffective. On the other hand, as far as the retention effect between two separate sessions is concerned, we could not find any meaningful statistical implication on the effect for all three error guidance modes out of the 2-week session of training. These findings can contribute to the improvement of VR technologies by ensuring they are designed to optimize human sound localization abilities.

## 1. Introduction

VR is a term used to describe an environment or situation that is created using computers and multimodal interface tools. Since it was first developed in 1960 [1], VR technology has made significant advancements in both hardware and software, and has had an impact on various fields of application. Over time, VR technology has evolved into augmented and mixed reality (XR) technology, which combines virtual images with the real world to create immersive experiences in a mixed virtual and real environment. With recent advancements in wearable devices, such as Microsoft’s HoloLens 2, these immersive experiences are becoming more accessible in our daily lives, with the aim of providing users with a fully immersive experience through multiple senses. One crucial aspect of creating a visually and aurally rich environment that can offer high-quality experiences to individuals, regardless of their visual and audio perception differences, is immersive audio technology. This technology creates an environment in which listeners feel completely surrounded by sound, adding a new dimension to videos, games, and music by delivering a highly realistic experience. In VR applications, the importance of immersive audio becomes even more evident, as it allows users to perceive sounds coming from all directions, enhancing their engagement with virtual or real-world scenarios.

The key to creating a more realistic and immersive experience lies in three-dimensional (3D) sound localization. This involves accurately identifying and mapping the sources of sounds, allowing users to perceive where each sound is coming from in space. In a VR environment, it is crucial to synchronize the 3D sound localization with the user’s visual experience. When the auditory cues align seamlessly with the visual cues, listeners not only feel more immersed, but also become more attuned to the VR encounter [2].

This study takes advantage of the capabilities of the HoloLens 2 to deliver a compelling auditory experience. The HoloLens 2 uses a head-related transfer function (HRTF) within its audio engine, which allows for a more immersive auditory experience. Unlike other VR devices that mainly focus on visual elements, the HoloLens 2 offers a unique mixed-reality feature that combines both visual and auditory effects. It also provides accurate positional and aural information. With the HoloLens 2, sounds can be delivered from different directions and distances in the virtual world, mirroring real-world parameters such as direction, height, distance, and frequency.

The HoloLens 2 utilizes our natural ability to hear sound in three dimensions by leveraging subtle cues like the time and intensity differences between our ears, which are represented by the head-related transfer function (HRTF). The processing of the HRTF is based on the distinct characteristics of each person’s ear, including its shape and size, which influence how sound is perceived when it enters the auditory canal. Although HRTF technology greatly enhances the immersive auditory experience by creating spatial realism, the individual variations in ear structure make the HRTF filters variable and necessitate personalized calibration.

The HoloLens 2 system comes with a default 5.1 multichannel HRTF. However, this general approach to using the HRTF may not provide an optimized experience for individuals with different HRTF perceptual characteristics. Research is currently being conducted to calculate and apply personalized HRTFs, but this field is still in its early stages [3,4,5,6,7].

In a study involving users of VR technology, researchers compared personalized HRTFs to generic ones [8]. The study found that, during a static localization test, there was no statistically significant difference in accuracy between personalized and generic HRTFs. Only a slight improvement was observed with personalized HRTFs. This suggests that, unless specific circumstances require it, personalized HRTFs may not be necessary for all VR users. Instead, those seeking a high level of auditory immersion are encouraged to use generic HRTFs. Our hypothesis is that, in addition to improving personalization and using motion-compensating HRTFs, it is important to train users to adapt to errors caused by individual differences and motion.

Several studies have conducted training sessions to help individuals adjust to common HRTFs. A recent study [9] revealed significant improvements in sound localization accuracy, specifically in reducing errors in spherical, lateral, and polar angles. Over the course of 3 days and a total training duration of 108 min, participants achieved error reductions ranging from 5% to 20% in all aspects of the experiment. However, the study focused solely on short-term effects, leaving the long-term effectiveness of VR-based sound localization training unexplored. Factors such as individual differences, familiarity with VR technology, and auditory processing abilities may also influence the outcomes, potentially limiting the broader applicability of the training.

In another study [10], participants underwent a four-week training program that involved variations in sound intensity along both horizontal and vertical axes. The target sound was positioned at the center of a circle with a radius of 6 m, covering positions 22.5° above and below the horizontal line. Initially, significant improvements were observed during the first two weeks of training. However, progress appeared to level off during the following two weeks. It is possible that the predetermined sequence of modules used in the study introduced a precedence effect, which may have influenced the results. Furthermore, using identical test modules during the evaluation phase could have introduced bias due to participants’ familiarity with the training materials.

Another study [11] investigated the enhancement of sound localization through a virtual hearing game. Rather than using traditional methods of location learning, researchers introduced a new training approach called the “Hoy-Pippi” 3D sound game. The participants were divided into two groups: a trained group and a non-trained group, each consisting of five participants. The results showed significant improvements in identifying the positions of sound sources in the trained group. However, a limitation of the study was that the stimuli azimuth only included seven sound source positions within 0–90° angles at 15° intervals. Therefore, a comprehensive examination of learning from different angles was not fully explored. These studies collectively highlight the evolving strategies for improving sound localization abilities in VR environments.

In this study, we sought to determine the most effective method for training users to adapt to the localization of sounds generated in VR. We looked at three training methods—Visual Guide 1, Visual Guide 2, and a Sound Guide—and evaluated the impact of each on improving 3D sound localization within a HoloLens-based VR environment. The experiment focused on observing how individuals adapted to common head-related transfer functions (HRTFs) through training. Comparing pre- and post-training, we found that the training method combined with kinesthetic feedback produced good results, but the impact on short-term retention over six sessions was minimal. We believe our findings can contribute to the development of VR experiences that incorporate user adaptation through training.

## 2. Literature Review

The role of VR in auditory training, specifically in the area of altered localization cues, has been gaining more attention in recent years [12]. The ability of the human auditory system to determine the source of sounds relies on a delicate balance between anatomical and environmental cues [13]. As technology advances, it becomes increasingly important to explore how these cues can be manipulated in a controlled environment such as VR.

In a study examining how auditory spatial perception adapts, researchers found that temporarily changing the shape of a listener’s ear affected their ability to locate sounds [14,15]. This discovery emphasizes the remarkable adaptability and flexibility of the human auditory system. Building on this idea, another group of researchers discovered that individuals can enhance the accuracy of sound localization by regularly exposing themselves to modified sounds [16]. This implies that with consistent and focused training, it is possible to improve an individual’s ability to locate sounds.

As the field of VR has advanced, studies have begun exploring its potential applications in auditory training [17]. One study focused on rapid adaptation to altered localization cues by immersing participants in a VR environment [18]. The results indicated that VR shows promise as a tool for training and adjusting auditory perception. Recognizing the interconnected nature of our senses, another research project investigated the integration of auditory and other sensory inputs, particularly vision [19,20]. Studies have shown that combining auditory cues with non-auditory cues can greatly enhance sound localization abilities. This multisensory approach emphasizes the importance of creating comprehensive training environments. Recent advancements in VR technology have also made it possible to convincingly simulate real-world auditory environments [21]. Studies that incorporated realistic simulations of room acoustics into VR have shown significant improvements in participants’ ability to localize sounds [22,23]. This body of evidence points to VR’s potential not only as a research tool but also as an effective training medium.

Several relevant studies have been conducted regarding VR and its impact on training. These studies considered various factors such as the use of visual helpers, individual HRTFs, and modified individual HRTFs. One notable study by Hofman et al. [24] investigated the neural calibration of sound in relation to spatial feedback from other sensorimotor systems. Their groundbreaking research revealed that altering the structure of the outer ear affected initial sound localization. However, individuals were able to relearn accurate sound localization over time, despite the modified ear structure. This adaptation did not disrupt the neural representation of the original cues, suggesting that the human auditory system is flexible and capable of recalibrating in response to changes in sound localization cues. Another study by Zahorik et al. [25] focused on auditory distance perception in both real and virtual environments. Their research demonstrated that the environmental context, whether real or virtual, significantly influenced how listeners judged distances based on auditory cues. This highlights the importance of realistic VR simulations in effective auditory training. Furthermore, Parseihian and Katz [26] conducted a study to explore VR’s potential for auditory spatial representation. They used VR to modify auditory localization cues and found that participants were able to adapt to changes within a virtual auditory space. This study laid the foundation for understanding the potential of VR in auditory training.

Building upon the concept of adaptability, Majdak et al. [27] conducted a study to investigate the impact of long-term training on sound localization. By exposing participants to spectrally altered sounds, they discovered that individuals could improve their ability to locate sounds with continued training. This suggests that individuals, including those with hearing impairments, have the potential to enhance their awareness of auditory space through training. Shifting the focus to auditory localization cues related to the head, Mendonça [28] provided a comprehensive review of adaptations in the auditory domain. The review encompassed strategies such as ear blocks, electronic hearing devices, and modified head-related transfer functions (HRTFs), all of which have been utilized to modify localization cues. Mendonça emphasized the rapid adaptability achieved through training, in contrast to the effects of mere exposure to sound. In line with these findings, Carlisle [29] reported on the plasticity of the adult auditory system, particularly in relation to altered spectral cues. Carlisle highlighted the interaction between auditory and non-auditory inputs and their influence on the perception of sound origin. The study revealed that even short-term training can significantly enhance the speed and extent of adaptation to altered auditory cues, suggesting a potential avenue for rehabilitative training with hearing aids and cochlear implants. Valzolgher et al. [30] conducted research on using multisensory training to help individuals relearn spatial hearing skills. They found that training adults with normal hearing using sound-oriented motor behavior could lead to generalized learning effects for various auditory spatial tasks. This suggests that a multimodal approach has the potential to improve sound localization abilities. Ahrens and colleagues [31] examined the integration of visual information with auditory perception in a VR context. Their study demonstrated that wearing head-mounted displays can influence the accuracy of sound localization. However, when individuals were provided with limited visual information, the localization error was reduced, highlighting the important relationship between auditory and visual cues in spatial perception. Lastly, Poirier-Quinot and Katz [32] focused on enhancing an individual’s adjustment to non-individualized HRTF profiles in a VR environment. They achieved accelerated improvements in audio localization performance by incorporating an interactive HRTF selection method, an active learning-based training program, and a simulation of room acoustics. Their study highlights the potential of VR and multimodal training to improve auditory adaptations. However, although many studies have examined auditory perception in virtual environments, there is still a significant gap in our understanding of how visual aids and individualized HRTFs together affect auditory training. Additionally, the impact of modifying individual HRTFs on auditory spatial perception in VR has been rarely explored. In light of these gaps, the goal of this study was to provide a more comprehensive examination of the role of visual aids, along with individual and modified HRTFs, in shaping auditory perception and training in VR environments. This will not only contribute to the existing literature but also offer practical insights for the development of effective auditory VR tools and therapies.

## 3. Experimental Design

### 3.1. Discrepancies in Sound Localization between HoloLens 2-Generated and Real Sound Sources

Previous studies have attempted to train individuals to localize sounds in real-world settings [33]. However, we hypothesized that there could be distinct variations between real-world auditory environments and those generated by a custom HoloLens application created using Unity. To explore this, we conducted an experiment with 8 participants. Our objective was to identify potential dissimilarities between real-world sounds and those simulated by the HoloLens system. The experiment was conducted in a controlled testbed where external noise was restricted to 10 dB. The testing room was 10 m long, 5 m wide, and 3 m high. Participants, seated and wearing HoloLens 2 headsets, participated in the experiments. Azimuth, distance, and altitude were measured with the position of the HoloLens 2 as the reference point. The test involved playing a 256 Hz tone for five seconds in five consecutive sessions at each of eight randomly selected locations, repeated a total of 40 times. Additionally, another experiment was conducted using real sound; speakers played the same 256 Hz tone from random locations mimicking the positions generated by the HoloLens 2, allowing for a comparison of the perceived differences between virtual and real sound sources.

Figure 1 compares the discrepancies between the HoloLens 2 and real-world sounds. It consists of 3 graphs that represent distance, azimuth, and elevation errors. The x-axis indicates the actual location of the sound source, while the y-axis shows the difference between the HoloLens 2 and real-world sounds at that specific position, with (a) indicating the error in distance in millimeters, (b) the angular error in azimuth, and (c) the angular error in elevation.

In terms of distance perception, participants consistently perceived the HoloLens 2-generated sound as being closer than the actual location of the real sound, regardless of the distance. When it came to azimuth estimation, both real and HoloLens 2 sounds tended to be localized to the right. However, there was a subtle difference: the inclination to localize towards the right was less pronounced for the HoloLens 2-generated sound compared to the real-world sound. Additionally, the sound was more prominently localized towards the right when the source was closer to the center. As for elevation perception, participants tended to perceive sounds as higher when they were situated at lower elevations, and vice versa. The HoloLens 2-generated sound consistently elicited lower elevation ratings compared to their actual positions. These distinct patterns of localization indicate discernible differences between real and HoloLens 2-generated sounds. Therefore, it is clear that sound localization training for real sound versus HoloLens 2-generated sound should differ due to their unique perceptual characteristics.

### 3.2. Training Human Subjects to Adapt to the Discrepancy

In our experiment, the researcher inputted the location coordinates, and the computer generated a sound based on them. The generated sound was then played through the HoloLens 2, and the participant listened to it. The participant used a stick to indicate the location of the sound. The experimenter compared the participant’s indicated location to the actual location of the sound and provided guidance. Figure 2 illustrates the sequence of the experiment.

The experimenter and participant collaborated in a controlled testbed environment, where external noise was limited to 10 dB. Before entering the testbed, the experimenter introduced the participant to the entire experiment and confirmed their gender, age, and hearing status. Once inside the testbed, the participant sat in a designated test chair, wore the HoloLens 2 headset, held a pointer in their hand, and faced straight ahead. During Visual Guide 1 and Visual Guide 2, the sound visualization feature of the HoloLens 2 was turned off so that the participant could not see the location of the sound. The location of the sound was only visible in the guided situation (Figure 3). In the Sound Guide, the subject wore an eye patch and listened to the sound generated by the HoloLens 2.

Participants went through 3 experimental sessions in sequence: a pre-training test, training session, and post-training test session. The test and training sessions included six spots, each with different elevations, distances, and azimuths.

During both the testing and training sessions, the experimenter played a 256 Hz tone five times for five seconds using the HoloLens 2. Participants listened to the sound and had to guess its source by pointing a pointer towards the location they believed the sound was coming from, taking into account the length of the stick. The experimenter then analyzed the coordinates where the participants pointed.

Each participant was instructed to stand at a series of points. At the first point, participants repeated the listening experience three times. Then, they moved to the second point for three more listening experiences. After that, they moved to point six, then returned to point one, and finally went back to point six for three more repetitions. Participants were guided to the actual location of the sound source using the HoloLens 2 in three different ways: Visual Guide 1, Visual Guide 2, and a Sound Guide. There was no separate training session. Once the experiment was completed, the participants and the experimenter left the testbed. The entire experiment took approximately 20–30 min per individual.

The locations of sound sources used for training were stored in a FLIR (Forward-Looking Infrared) camera system from Vicon, while the predicted location was specified using X, Y, and Z spatial coordinates. To ensure accurate analysis, the training point was then transformed into a spherical coordinate system (r, θ, φ). The transformed values of the training point and the location predicted by the subject were calculated using the following formula for absolute error:
(1)$$E_{v}=|V_{a}-V_{o}|$$
where $V_{o}$ is the point location predicted by the subject and $E_{v}\;$ is the absolute error.

Twelve subjects of various genders and ages participated in the experiment. The average age was 23.4 years. None of the subjects had any prior experience using a HoloLens 2 (Table 1).

### 3.3. Experimental Setup

In order to obtain 3D spatial coordinates and measure the positions of both the sound source and the subject’s predicted position, we utilized a Vicon camera system. Beacon sensors were employed to predict the location of the sound source. The Vicon camera system comprised four infrared cameras that captured measurements within a 3 m wide, 2 m long, and 2 m high 3D space. The HoloLens 2, connected to the Vicon camera system, utilized the HRTF to simulate sounds originating from various directions and distances in the virtual world.

In this experiment, participants were given instructions to learn locations by triggering a buzzer at 5sec intervals. This prompted the experimenter to generate sounds from 6 different locations. The participants listened to the sounds coming from the HoloLens 2 and directed a pointer towards the location they believed the sound was coming from. A beacon sensor was attached to the far end of the pointer. Once the participants determined the location of the sound source, the coordinates recognized by the beacon sensor were sent to the experimenter. The experimenter then compared the coordinates with the actual location of the sound source, checking for errors in distance (r), azimuth (θ), and elevation (φ). The experimental space was a semicircle with a maximum radius of 1.5 m and a maximum height of 1.3 m, relative to the participant’s location.

In Figure 3, (a) is the actual space of the testbed under experimentation, and (b) is an illustration of the testbed space using the guides in Visual Guide 1 and Visual Guide 2.

### 3.4. Experimental Procedure

As described in Section 3.2, the experiment included a pre-training test session, followed by a training session, and concluded with a post-training test session. In the pre-and post-training sessions, we obtained subjects’ performance without supplying feedback about their training, whereas in the training sessions, we provided feedback on their training immediately after each session. For comparison purposes, we considered 3 types of feedback: Visual Guide 1, Visual Guide 2, and a Sound Guide, as shown in Figure 4. Visual Guide 1 provided visual feedback on the location of the sound source when the subjects lowered their arm after they finished pointing, while Visual Guide 2 provided visual feedback on the location of the sound source immediately after the subjects finished pointing, and the experimenter ended the training session by pointing to the red box shown in the visual feedback. In this case, we used visual and kinesthetic feedback. Figure 5 depicts the feedback about the location of the actual sound in Visual Guide 1 and Visual Guide 2. The red boxes are 3 cm × 3 cm × 3 cm cubes. 

The Sound Guide was designed to provide a guiding sound, enabling the subject to move the pointing stick along the guide sound and reach the actual direction of the sound. The pitch of the guide sound would increase or decrease depending on the distance. Once the pointing stick reached the location of the sound, the sound would stop and the feedback from the Sound Guide would cease. Please refer to Figure 4 for a flowchart illustrating each guide.

## 4. Results

### Testing the Effect of the Six-Session Training of Sound Source Locations

Figure 6 features a graph that outlines the pre-training performance across six training sessions. The observed reductions and increases in error were determined by comparing the results between each nth and nth + 1 session and subsequently averaging these changes across all sessions. For Visual Guide 1, there was a decrease of 6% in elevation error, a reduction of 3.3% in azimuth, and an increase of 4.4% in distance. For Visual Guide 2, we noted a decrease of 3.8% in elevation error, a slight increase of 0.4% in azimuth, and a decrease of 6.7% in distance. Lastly, the Sound Guide exhibited an increase of 1.1% in elevation error, a rise of 2.7% in azimuth, and a decrease of 2.7% in distance. These results elucidate the variances in performance modifications across the training sessions.

Figure 7 illustrates the post-training performance, with results similarly analyzed by comparing each nth and nth + 1 session and averaging the changes across all sessions. For Visual Guide 1, there was a 6.1% decrease in elevation error, a 3% reduction in azimuth, and a 2.2% increase in distance. Visual Guide 2 showed a 6.2% decrease in elevation error, a 6.3% reduction in azimuth, and a significant 16.4% decrease in distance. Meanwhile, the Sound Guide exhibited a 1.7% decrease in elevation error, a 2.9% increase in azimuth, and a 6.1% decrease in distance.

Figure 8, Figure 9 and Figure 10 display the pre- and post-training results for Visual Guide 1, Visual Guide 2, and the Sound Guide, respectively, highlighting the training effects observed from day 1 to day 6. In terms of elevation, the Sound Guide decreased errors by 5.3% and Visual Guide 2 reduced errors by 7.9%, whereas Visual Guide 1 unexpectedly increased errors by 7.1%. For distance, both the Sound Guide and Visual Guide 2 showed substantial training effects, reducing errors by 14.7% and 14.9% respectively, whereas Visual Guide 1 exhibited no notable error change. Regarding azimuth, all three training methods demonstrated significant improvements; Sound Guide errors decreased by 9.7%, Visual Guide 1 by 5.8%, and Visual Guide 2 by 15.1%. Overall, Visual Guide 2 showed the most significant training effect, although the Sound Guide also presented considerable efficacy, underlining the importance of tailored training approaches to enhance spatial audio accuracy across different dimensions. 

We observed a significant increase in elevation errors during the third session of the Sound Guide. This can be attributed to the proximity of objects, which often leads to confusion regarding elevation and results in participants guessing and randomly selecting positions without certainty. Specifically, on one particular day, this issue was notably evident in two out of four participants, highlighting considerable individual variability in sensitivity to elevation. Rather than dismissing these observations as outliers, we included them in our dataset as they exemplify a unique characteristic of perceptual response to elevation, underscoring that sensitivity to elevation can vary significantly among individuals.

## 5. Analysis

We analyzed the statistical significance of the experimental results described in Section 4 in terms of the training effect associated with the three training modes on 3D sound source localization. When analyzing the training effect, we considered (1) the training effect associated with individual sessions, which refers to the improvement of localization performance associated with different training modes within a training session, and (2) the effect of retaining trained performance between sessions, which refers to the temporal retention of the training effect between subsequent sessions associated with the three different training modes. In terms of statistical analysis, we took the following two factors into consideration: (1) since the same individuals were involved in analyzing the training effect between pre- and post-training within a session, as well as the retention effect between subsequent sessions, we used paired statistics, and (2) we defined a measure of performance improvement in terms of the individual changes in the statistical distribution of localization. For the latter, we considered both the means and standard deviations (SDs) associated with the two localization datasets from each individual. To this end, we adopted the following SD-compensated mean-difference distance, ***d***, as a paired measure of performance improvement for individuals.
(2)$$d=\frac{\mu_{1}-\mu_{2}}{\sqrt{{{\sigma_{1}}^{2}+\sigma_{2}}^{2}}}$$

$\mu_{1},\sigma_{1}$: the mean and standard deviation of the localization errors by an individual at a particular session. $\mu_{1}\;and\;\sigma_{1}$ serve as the reference for measuring training and retention effects. 

$\mu_{2}$, $\sigma_{2}$: the mean and standard deviation of the localization errors by the same individual at the same session as those of $\mu_{1}$. $\mu_{2}$ and $\sigma_{2}$ are evaluated in terms of $\mu_{1}\;and\;\sigma_{1}$ to measure the improvement.

There are other measures such as the Mahalanobis distance [34] and KL divergence [35] available for use to define the distance between two statistical distributions instead of d as defined in Equation (2). Note that d is similar to the t-value used for the paired *t*-test except that $\mu_{1}-\mu_{2}$ in ***d*** is a signed number to represent the increase/decrease in performance, while no sample size is incorporated into $\sigma_{1}\;and\;\sigma_{2}$. 

### 5.1. Statistical Significance of Training Effect Associated with the Performance Improvement from Individual Training Sessions

Table 2 summarizes a comparative analysis of the effectiveness of the three different training modes on sound localization. The mean ***d***-value represents the average of all the ***d***-values from four individuals and six training sessions. The ***d***-value of an individual indicates the difference in statistical distributions between their pre- and post-training localization errors. For a visual comparison, Figure 11 displays the mean ***d***-value associated with the three training modes.

Table 2 and Figure 11 indicate that, when it comes to the training effect associated with individual sessions, Visual Guide 2 and the Sound Guide clearly demonstrate their effectiveness, whereas Visual Guide 1 shows its limitations. We believe that this result is due to the use of kinesthetic/postural feedback in Visual Guide 2 and Sound Guide modes, which is not provided in Visual Guide 1. We hypothesize that the combination of visual-kinesthetic/posture or sound-kinesthetic/posture multimodal coordinate stimulus enhances perceptual effectiveness for localization learning. Additionally, we observed that, as shown in Figure 11, training sound localization in terms of azimuth and distance is much more effective than training elevation, regardless of the training mode. Note that, when wearing the HoloLens, speakers are often positioned above the ears. This above the ear position of speakers not only makes it harder for the HoloLens to generate its sound to emulate the human HRTF (head-related transfer function) associated with elevation in good quality but also introduces a difficulty in handling the sound reflection over the shoulder, which is crucial for localizing the elevation of a sound source. To conduct a quantitative analysis on the impact of the three different training methods on sound localization performance, we carried out hypothesis test H1. This test aimed to measure the confidence level associated with the statistical improvement in sound localization between the pre- and post-training sessions.

**H_1_** **(mode):**
*The mean **d**-value indicates the improvement in sound localization performance for the given training mode.*


Since the mean ***d***-value was computed using individually paired statistics, the H_1_ (mode) test followed a one-sample *t*-test based on the standard deviations (SDs) presented in Table 2. The results of the one-sample *t*-test, which were used to determine H_1_ (mode), along with the corresponding confidence levels for the three distinct training modes, are summarized in Table 3.

Table 3 shows that, in terms of the H_1_ (mode) test of the training effect per session, Visual Guide 2 and the Sound Guide both demonstrate a significant effect with a high confidence level of over 99% for sound localization in azimuth and distance. However, the effect in elevation is less significant. In contrast, Visual Guide 1 has some effect in azimuth, with a confidence level of 87.7%, but no effect at all in elevation and distance.

### 5.2. Statistical Significance of Training Effect Associated with Performance Improvement between Training Sessions

Similar to the way the effect of training sessions is evaluated by measuring the statistical differences in localization errors between pre- and post-training sessions based on mean ***d***-values, we also examined the possible retention of training effect between adjacent sessions by measuring the mean ***d***-values obtained by averaging ***d***-values from individuals. In examining the retention of training effect, we considered the mean ***d***-values associated with pre-training statistics between sessions and the mean ***d***-values associated with post-training statistics between sessions.

#### 5.2.1. Retention Effect between Adjacent Pre-Training Sessions

Table 4 summarizes the average of all ***d***-values obtained from four individuals and six adjacent pre-training sessions. The ***d***-values represent the difference in the statistical distribution of sound localization errors between adjacent pre-training sessions. Figure 12 visually presents the mean ***d***-value for the three training modes, allowing for easy comparison. Both Table 4 and Figure 12 indicate that, unlike the session-wise training effect, the retention of the session-wise training approach between adjacent pre-training sessions does not show any effect for all three distinct training modes or for any of the three localization axes. This lack of retention is clearly indicated by the H_1_ (mode) test, as summarized in Table 5, where no test item shows a confidence level of over 90%. In conclusion, while there are some minor improvements in specific spatial dimensions when using different guides, such as distance training with Visual Guide 2, the overall impact remains statistically insignificant.

#### 5.2.2. Retention Effect between Adjacent Post-Training Sessions

Table 6 summarizes the average ***d***-values obtained from four individuals and six adjacent post-training sessions. For comparison, Figure 13 visualizes the mean ***d***-value for each of the three training modes. Similar to the retention effect observed between adjacent pre-training sessions, there is no significant retention effect across adjacent post-training sessions, except for the case of distance localization training by Visual Guide 2. Table 7 shows that only distance localization by Visual Guide 2 has a confidence level of 99%, which is higher than 90%. It is worth noting that, in terms of retention effect, Visual Guide 2 performs better than the Sound Guide, suggesting that visual perception may have a stronger influence on retaining the training effect than kinesthetic/postural perception. Furthermore, we observed that both Visual Guide 2 and Visual Guide 1 demonstrate a higher level of confidence in retaining the training effect during post-training sessions compared to pre-training sessions. We hypothesize that visual guidance is more effective in retaining training efficacy, rather than promoting memorization.

## 6. Discussion

We chose to use a pointing stick for the user to express their perceived sound locations as it represents the most accurate way to do so. However, it is expected that the statistics involved in the pointing error itself may influence those of sound localization errors. Based on the high accuracy in pointing together with the fact that pointing accuracy affects all the experiments under comparison with the same statistics, we adopt the pointing error as it is for the sound localization error.

To assess the effectiveness of the sample size used for the statistical analysis in Section 5, we conducted a sample size effectiveness evaluation. This evaluation aimed to determine if the sample medians of ***d***-values, obtained from three different modes of localization experiments, fell within their respective 95% confidence intervals (L95 and U95) derived from the sample average and standard deviation. Table 8, Table 9 and Table 10 summarize the results, indicating that all cases of statistical analysis for the three distinct training modes and three different localization axes met the L95-U95 criteria. These findings confirm the effectiveness of the sample sizes we collected. 

In our study, we divided twelve participants into three groups, with each mode comprising four subjects. We conducted an evaluation using data from a single session to validate the dataset size. This approach allowed us to uniformly assess the validity of the dataset sizes across all sessions. The results demonstrated that in 92.6% of the cases, the median values fell within the 95% confidence interval across all three modes—azimuth, distance, and elevation—across six sessions (54 datasets in total). Only 7.4% of the cases met the less stringent 90% confidence interval. This analysis confirmed that our data hold sufficient reliability.

Note that our study highlights the critical need for technology, such as the HoloLens, to be designed with an understanding of user interaction and adaptation requirements. We discovered that training approaches for sound localization vary significantly between elevation, azimuth, and distance, indicating a need for tailored design strategies to enhance user adaptation, especially for elevation. This study underscores that while technology cannot completely adapt to the user without effort, informed training methods can significantly ease the adaptation process, guiding the development of more intuitive and user-friendly technological tools. Building on these insights, our findings can inform specific design improvements for the HoloLens to better adapt to users, as follows. (1) We found that the HoloLens 2 users mostly failed to adapt to the sound localization in terms of elevation. That is, as far as elevation localization is concerned, no tangible training effect was observed for all three modes of training: Visual Guide 1, Visual Guide 2, and the Sound Guide. This suggests that the current spatial audio system of the HoloLens 2 may not provide sufficient cues for accurate elevation perception and need improvement in its design. (2) We recognized the need to train HoloLens 2 users somehow each time when they started to use the device, so that they could adapt to sound localization with sufficient accuracy. This is based on our finding that the user performance in sound localization was not satisfactory when they started to use the device during the two weeks of training sessions designed for observing the temporal retention of training effect. And, no statistically significant temporal retention of the training effect was observed during the two weeks of training sessions. Therefore, we consider it highly recommendable for the device to include a training program that can help the user quickly adapt to the device for accurate sound localization. As for the proposed training program to promote user adaption, our finding on the impact of kinesthetic/postural feedback on the training effect, i.e., that the Visual Guide 2 and Sound Guide modes of training with kinesthetic/postural feedback were much more effective than Visual Guide 1 in localization performance, may provide an essential guideline for implementation. We hope that our study provides a foundation for future research and development aimed at improving the design of the HoloLens, ensuring it meets the diverse needs of its users and enhances their overall experience.

## 7. Conclusions

In this study, we conducted an experimental investigation to examine how training impacts the accuracy of HoloLens-based sound localization. The experiment compared three different training modes: Visual Guide 1, Visual Guide 2, and the Sound Guide. Additionally, we compared the effects of training on three different localization axes: elevation, azimuth, and distance. Specifically, the Visual Guide 2 and Sound Guide modes were designed to be multimodal, incorporating kinesthetic/postural perception with visual and audio perception, respectively. This was done to evaluate the influence of kinesthetic/postural perception on sound localization. We collected pre- and post-training data over six sessions, spanning a two-week period, for each of the three training modes. There were four subjects per training mode, resulting in a total of 18 pre-training and 18 post-training paired data points per subject from each session. We analyzed the collected data by assessing pairwise statistical changes represented by ***d***-values, focusing on both the session-wise training effect and the between-session retention effect. Our findings indicate that Visual Guide 2 and Sound Guide modes significantly improve localization performance, particularly in terms of azimuth and distance. However, the effectiveness of these modes appears to be significantly reduced when it comes to elevation. This can be attributed to the lower sensitivity of human head-related transfer function (HRTF) characteristics in elevation compared to azimuth or distance, as well as the positioning of the HoloLens speakers above the ears. On the other hand, we did not observe a significant between-session retention effect with a satisfactory level of confidence. This suggests that the impact on short-term retention across the six sessions is marginal. However, we did notice that visual perception seems to play a more significant role in retention compared to sound perception. This observation highlights the need for further investigation into the long-term retention effects of HoloLens-based sound localization training. Going forward, our future plans include exploring the long-term effects of these training modalities, particularly in relation to retention effects, and investigating the possibility of detecting training and retention effects through changes in brain signals using functional magnetic resonance imaging (fMRI).

## Figures and Tables

**Figure 1 sensors-24-03442-f001:**
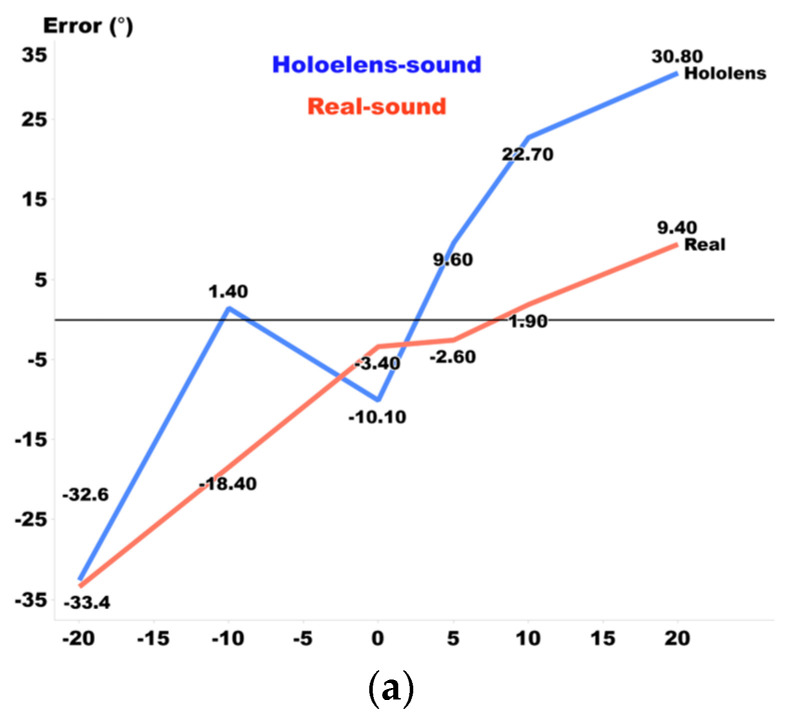
(**a**) Elevation, (**b**) azimuth, and (**c**) distance error between HoloLens sound and real sound.

**Figure 2 sensors-24-03442-f002:**
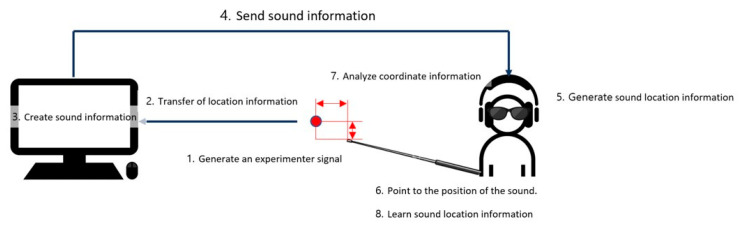
Experimental sequence.

**Figure 3 sensors-24-03442-f003:**
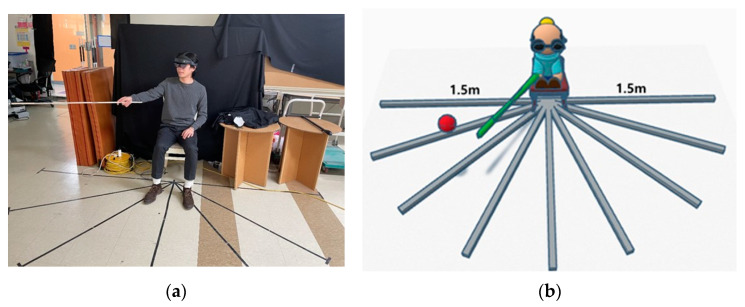
(**a**) Photograph of the setup. (**b**) Predefined experimental space based on subject frontal arc of −90° to +90°, a maximum distance of 1.5 m, and a height of up to 1.3 m.

**Figure 4 sensors-24-03442-f004:**
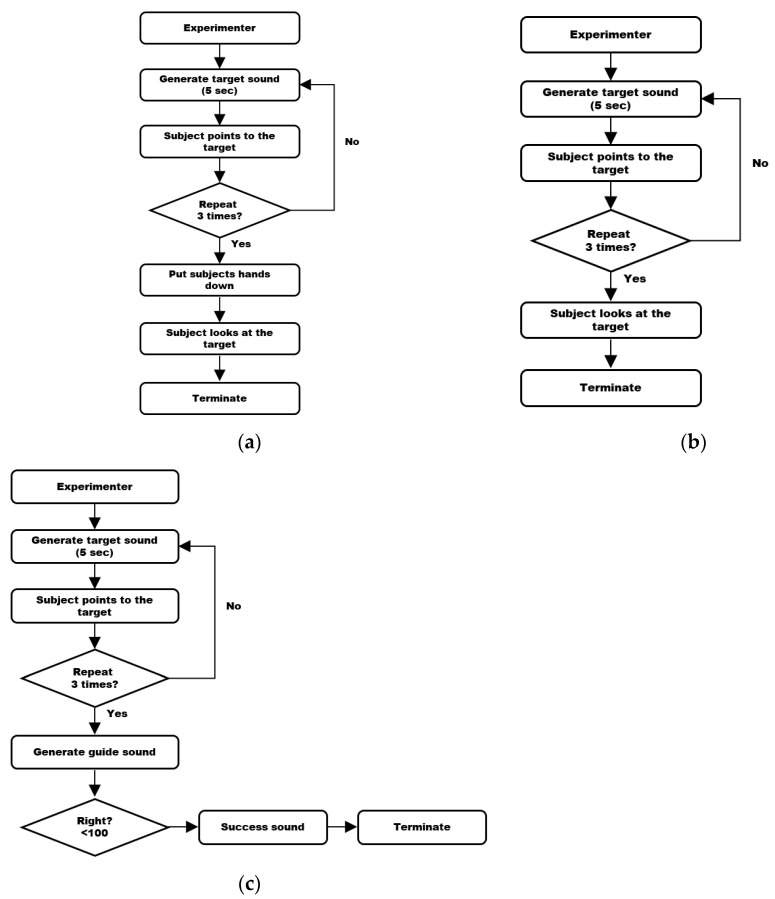
Flowcharts of (**a**) Visual Guide 1, (**b**) Visual Guide 2, and (**c**) the Sound Guide.

**Figure 5 sensors-24-03442-f005:**
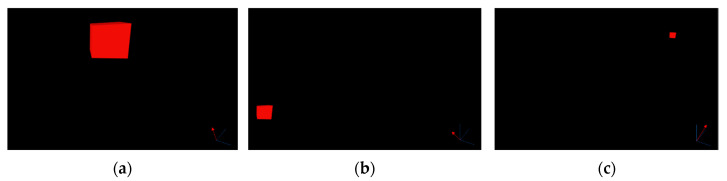
(**a**) Example of visual feedback used in Visual Guide 1 and Visual Guide 2 (3 cm × 3 cm × 3 cm red cube) and a comparison of distance, height, and azimuth based on the position of the red cube. (**a**) Elevation: 14°, azimuth: −14°, distance: 412 mm; (**b**) elevation: −45°, azimuth: −51°, distance: 1510 mm; (**c**) elevation: 16°, azimuth: 32°, distance: 1711 mm.

**Figure 6 sensors-24-03442-f006:**
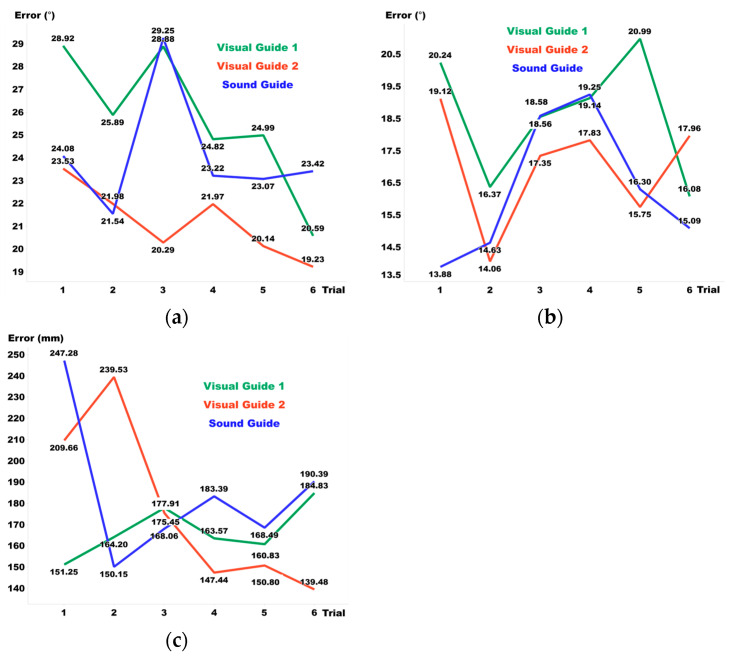
(**a**) Elevation, (**b**) azimuth, and (**c**) distance errors pre-training.

**Figure 7 sensors-24-03442-f007:**
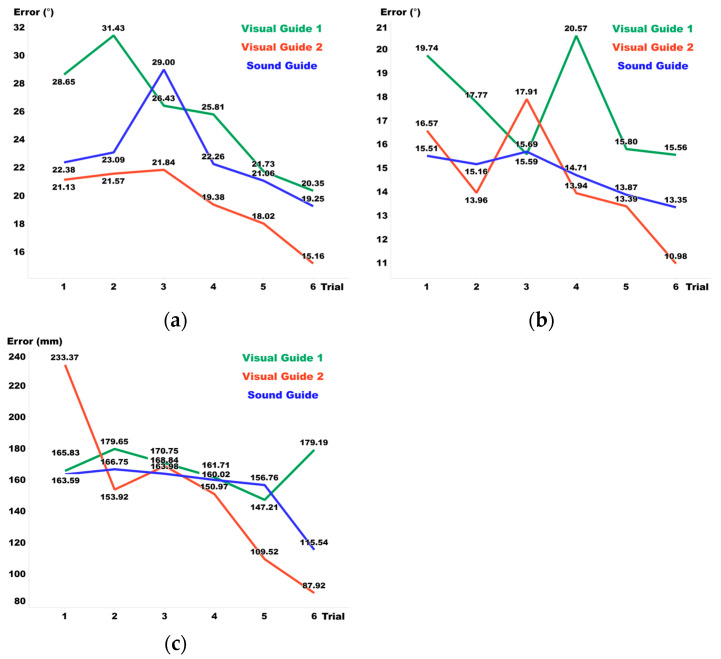
(**a**) Elevation, (**b**) azimuth, and (**c**) distance errors post-training.

**Figure 8 sensors-24-03442-f008:**
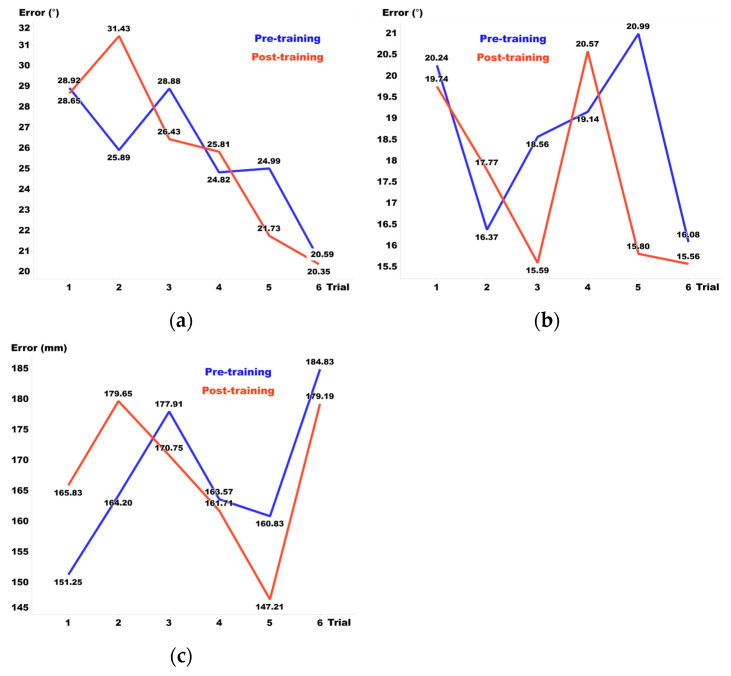
(**a**) Elevation, (**b**) azimuth, and (**c**) distance error before and after each day’s training in Visual Guide 1.

**Figure 9 sensors-24-03442-f009:**
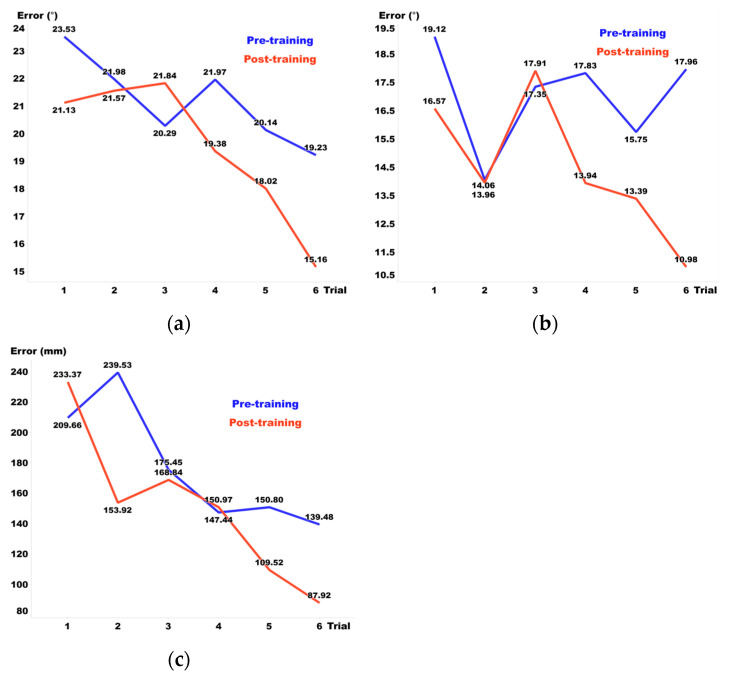
(**a**) Elevation, (**b**) azimuth, and (**c**) distance errors before and after each day’s training in Visual Guide 2.

**Figure 10 sensors-24-03442-f010:**
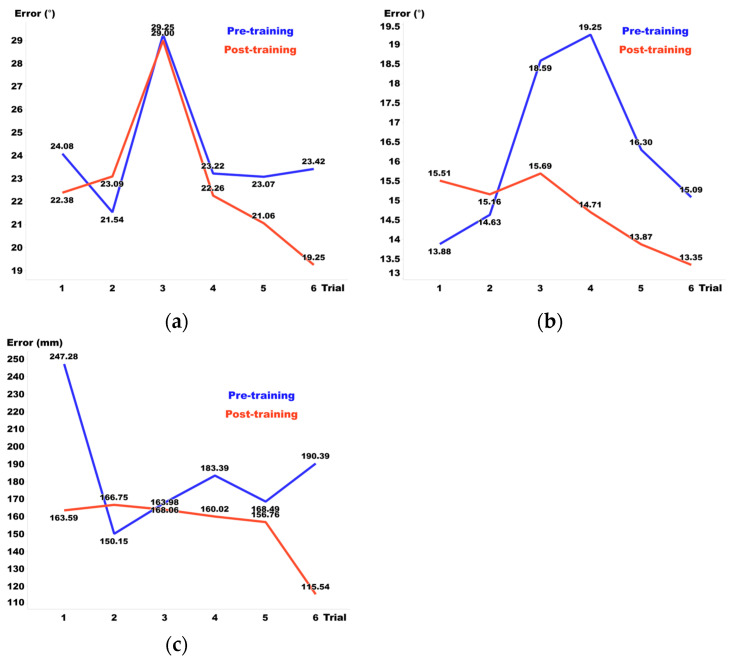
(**a**) Elevation, (**b**) azimuth, and (**c**) distance errors before and after each day’s training in the Sound Guide.

**Figure 11 sensors-24-03442-f011:**
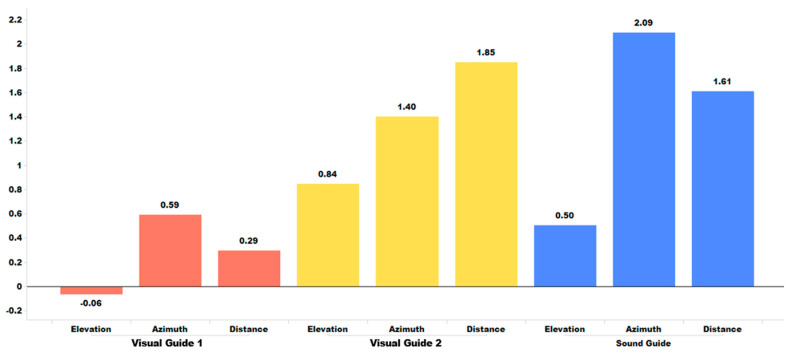
Pre- and post-training paired comparison of error statistics measured by ***d***-value.

**Figure 12 sensors-24-03442-f012:**
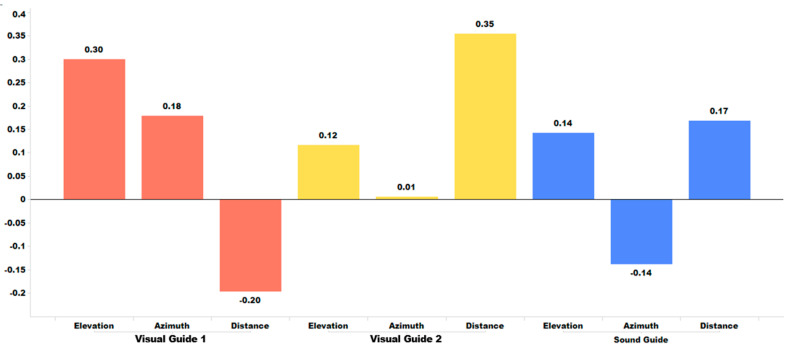
Mean ***d***-values between pre-training sessions for three distinct training modes.

**Figure 13 sensors-24-03442-f013:**
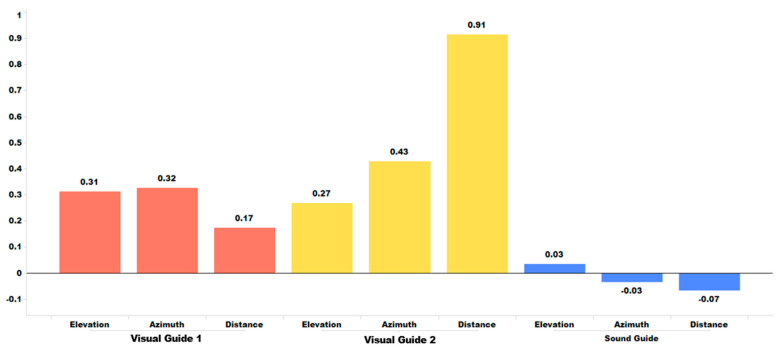
Mean ***d***-values between post-training sessions for three distinct training modes.

**Table 1 sensors-24-03442-t001:** Demographic information about the participants.

Training Activity Type	Participant	Sex	Age
Visual Guide 1	1	M	23
	2	F	21
	3	M	22
	4	M	25
Visual Guide 2	5	M	24
	6	F	21
	7	F	23
	8	M	26
Sound Guide	9	F	26
	10	M	22
	11	F	20
	12	M	29

**Table 2 sensors-24-03442-t002:** Pre- and post-training paired comparison of error statistics measured by mean ***d***-value.

Training Mode	Elevation		Azimuth		Distance	
	Mean *d*-Value	SD	Mean *d*-Value	SD	Mean *d*-Value	SD
Visual Guide 1	−0.062	0.287	0.5877	0.7932	0.2937	0.3457
Visual Guide 2	0.8424	1.132	1.4015	1.7381	1.8501	1.9921
Sound Guide	0.5028	0.7231	2.093	2.743	1.611	1.8273

**Table 3 sensors-24-03442-t003:** Results of the one-sample *t*-test for pre- and post-training comparison.

Training Mode	Elevation		Azimuth		Distance	
	*p*-Value	Confidence Level (%)	*p*-Value	Confidence Level (%)	*p*-Value	Confidence Level (%)
Visual Guide 1	N/A	negative	0.123	87.7	0.978	2.2
Visual Guide 2	0.115	88.5	0.000	99.9	0.000	99.9
Sound Guide	0.484	51.6	0.002	99.8	0.004	99.6

**Table 4 sensors-24-03442-t004:** Mean ***d***-values between pre-training sessions for three distinct training modes.

Training Mode	Elevation		Azimuth		Distance	
	Mean *d*-Value	SD	Mean *d*-Value	SD	Mean *d*-Value	SD
Visual Guide 1	0.3001	0.2071	0.1791	0.1351	−0.196	−0.248
Visual Guide 2	0.1168	0.0238	0.005	−0.088	0.3543	0.2573
Sound Guide	0.1419	0.299	−0.1374	0.3115	0.1686	0.2436

**Table 5 sensors-24-03442-t005:** Results of the one-sample *t*-test for evaluating the retention between adjacent pre-training sessions.

Training Mode	Elevation		Azimuth		Distance	
	*p*-Value	Confidence Level (%)	*p*-Value	Confidence Level (%)	*p*-Value	Confidence Level (%)
Visual Guide 1	0.227	77.3	0.310	69	N/A	negative
Visual Guide 2	0.496	50.4	0.765	23.5	0.104	89.6
Sound Guide	0.956	4.4	N/A	negative	0.686	31.4

**Table 6 sensors-24-03442-t006:** Mean ***d***-values between post-training sessions for three distinct training modes.

Training Mode	Elevation		Azimuth		Distance	
	*d*-Value	SD	*d*-Value	SD	*d*-Value	SD
Visual Guide 1	0.3116	0.4046	0.3249	0.3689	0.1734	0.2254
Visual Guide 2	0.2668	0.3598	0.4284	0.5214	0.914	1.011
Sound Guide	0.0344	−0.1227	−0.0326	−0.4815	−0.0658	−0.140

**Table 7 sensors-24-03442-t007:** Results of the one-sample *t*-test for evaluating the retention between adjacent post-training sessions.

Training Mode	Elevation		Azimuth		Distance	
	*p*-Value	Confidence Level (%)	*p*-Value	Confidence Level (%)	*p*-Value	Confidence Level (%)
Visual Guide 1	0.224	77.6	0.258	74.2	0.825	17.5
Visual Guide 2	0.260	74	0.121	87.9	0.000	99.9
Sound Guide	0.973	2.7	0.586	41.4	0.799	20.1

**Table 8 sensors-24-03442-t008:** Comparison of elevation data pre- and post-training.

	Visual Guide 1	Visual Guide 2	Sound Guide
Count	432	432	432
Median	−0.062	0.7484	0.4868
L95	−0.089	0.7356	0.4346
U95	−0.034	0.9491	0.5709
Average	−0.062	0.8424	0.5028
StdDev	0.287	1.132	0.7231

**Table 9 sensors-24-03442-t009:** Comparison of azimuth data pre- and post-training.

	Visual Guide 1	Visual Guide 2	Sound Guide
Count	432	432	432
Median	0.5557	1.3075	1.890
L95	0.4929	1.2375	1.833
U95	0.6424	1.5654	2.352
Average	0.5877	1.4015	2.093
StdDev	0.7932	1.7381	2.743

**Table 10 sensors-24-03442-t010:** Comparison of distance data pre- and post-training.

	Visual Guide 1	Visual Guide 2	Sound Guide
Count	432	432	432
Median	0.3017	1.8701	1.627
L95	0.2611	1.6624	1.4386
U95	0.3262	2.0379	1.7833
Average	0.2937	1.8501	1.611
StdDev	0.3457	1.9921	1.8273

## Data Availability

Data are contained within the article.

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
