# Peer review of "The Effect of Training on Localizing HoloLens-Generated 3D Sound Sources"

_sensors, 2024, doi:10.3390/s24113442_

Round 1

Reviewer 1 Report

Comments and Suggestions for Authors

Please find my review in the attached document.

Author Response

Attached, please find our detailed response to your comments on our manuscript, including the requested revisions and additional information. We greatly appreciate your insightful feedback, which has significantly contributed to improving the quality of our work.

Thank you once again for your valuable input. We look forward to your feedback on the revised manuscript.

Reviewer 2 Report

Comments and Suggestions for Authors

The authors present three training modes to aid users use HRTF-based spatial audio reproduction.

The are several important issues with the manuscript:

- Passing on the work to the user to make the spatial audio system work seems counter-intuitive to the philosophy of "the technology needs to adapt to the user, not the other way around". Moreover, is the training going to impact the user's ability to hear in the natural world?

- Although it is expected, the phrase (lines 58-60) "When the auditory cues align seamlessly with the visual cues, listeners not only feel more immersed, but also become more attuned to the VR encounter." needs a reference to back it up.

- In section 3.1, the authors need to provide details of the acoustic setting on how the "real sound" was reproduced: size of the room, reverberation time, what was used as the sound source (was it a speaker reproducing a sine wave? or a speech recording? or was it an actual person?), how were the azimuth and elevation measured, what was their frame of reference, etc. In addition, how was the head positioning controlled? This is important since both the azimuth and elevation are dependent on it.

- Figure 1 needs to provide the error in a scale that corresponds to the variable. Meaning, for distance, in milimeters or meters (not "error percentage"); for angles, in radians or degrees. And, it seems that "elevation" is really "height", given that the error is given in milimeters. If so, it should be an angle instead.

- How was the error of the sound localization separated from the error of the pointer?

- It is unclear why the authors state that (lines 354-355) "The average elevation values before training, from sessions 1 to 6, remained unchanged for visual guide 1". In Figure 4.1a it is clear that the error went down for visual guide 1. Did the authors mean that the average error remained unchanged for sound guide? Since it is the one in which the average error in the first trial and the last trial are very similar. In the same manner, the authors state that (lines 356-358) "In terms of the azimuth angle [...] the sound guide decreased by 5.8%" while it clearly went up from the first and final trial.

- In fact, the explanations of the results shown in Figures 6, 7, 8, 9, and 10 are very confusing overall. How are the reduction percentages calculated? Why does the error in sound guide increased in the third trial before and during training?

- The axis font in all of the sub-figures in Figures 6, 7, 8, 9, and 10 needs to be increased.

- Considering that only 4 subjects were used per each training mode, this reviewer is not convinced that the statiscal analysis for each training mode is reliable. Even more so considering what the authors themselves stated for the study in reference [8], which was deemd incomplete (lines 95-96) "the study had a relatively small sample size, with only 7 to 11 participants per group". The sample size analysis discussed in Section 6 only establishes the size of the samples obtained from each subject, which do meet the L95-U95 criteria. However, it does not establish if the number of subjects does as well.

- Finally, the marginality of the results from the training sessions, in conjunction with the small amount of subjects per each training mode, does not say much more than what the other similar studies referenced by the authors have already stated.

Comments on the Quality of English Language

- There is a considerable amount of typos. Here is an incomplete list of the errors this reviewer found just in the second page of the manuscript:

. line 61: "compel-ling" -> "compelling"

. line 65: "ac-curate" -> "accurate"

. line 69: "lever-aging" -> "leveraging"

. line 70: "rep-re-" -> "repre-"

Author Response

(The authors gave the same response as above.)

Round 2

Reviewer 2 Report

Comments and Suggestions for Authors

Although the authors acknowledged and fixed several of this reviewers comments, there are still essential ones that haven't been:

- The authors believe that the number of subjects are enough, and have provided proof to such effect. Therefore, the authors should remove any criticism of other studies that are based on the number of subjects, since some of such studies use more subjects per trial than the authors did.

- This reviewer appreciates the added text about the philosophy of "the technology needs to adapt to the user, not the other way around", and the manuscript's findings are very interesting. However, the focus of this article shouldn't be "how to train the user to better adapt to the HoloLens", but "how can these findings inform the design of the HoloLens so that it better adapts to the user". To this effect, the authors should integrate their findings with the ones from the other referenced studies and provide recommendations of where the HoloLens is failing the user, and how it can be improved.

Author Response

(The authors gave the same response as above.)
